# Establishment of *Neurospora crassa* as a model organism for fungal virology

Shinji Honda[1], Ana Eusebio-Cope[2], Shuhei Miyashita[3], Ayumi Yokoyama[1], Annisa Aulia[2], Sabitree Shahi [2], Hideki Kondo[2] & Nobuhiro Suzuki [2✉]

The filamentous fungus *Neurospora crassa* is used as a model organism for genetics, developmental biology and molecular biology. Remarkably, it is not known to host or to be susceptible to infection with any viruses. Here, we identify diverse RNA viruses in *N. crassa* and other *Neurospora* species, and show that *N. crassa* supports the replication of these viruses as well as some viruses from other fungi. Several encapsidated double-stranded RNA viruses and capsid-less positive-sense single-stranded RNA viruses can be experimentally introduced into *N. crassa* protoplasts or spheroplasts. This allowed us to examine viral replication and RNAi-mediated antiviral responses in this organism. We show that viral infection upregulates the transcription of RNAi components, and that Dicer proteins (DCL-1, DCL-2) and an Argonaute (QDE-2) participate in suppression of viral replication. Our study thus establishes *N. crassa* as a model system for the study of host-virus interactions.

[1] Faculty of Medical Sciences, University of Fukui, Fukui 910-1193, Japan. [2] Institute of Plant Science and Resources, Okayama University, Kurashiki, Okayama 710-0046, Japan. [3] Graduate School of Agricultural Science, Tohoku University, 468-1, Aramaki-Aza- Aoba, Sendai 980-0845, Japan. ✉email: nsuzuki@okayama-u.ac.jp

Fungal viruses or mycoviruses are omnipresent in all major groups of fungi, and the majority of them show asymptomatic and frequently mixed infections[1–3]. Like viruses of other host kingdoms, fungal viruses have different types of genomes, although they largely are double-stranded (ds) and single-stranded (ss) positive-sense (+) RNA viruses[1]. Mycoviruses are increasingly discovered in fungi by the conventional detection of dsRNAs, a sign of RNA virus infection[4], via high-throughput sequencing of transcripts and/or dsRNA[5–7], and in silico mining of publicly available transcriptomic data[8,9]. Fungal viruses commonly have no extracellular route for entry with the exception of an encapsidated ssDNA virus[10], and are generally difficult to experimentally introduce into their host fungi extracellularly, which often impedes the progress of this kind of study. Only for some mycoviruses have protocols been developed for the experimental introduction or inoculation. Virion transfection is frequently used for encapsidated dsRNA viruses such as reoviruses[11], totiviruses[12], megabirnaviruses[13], and partitiviruses[14]. For several capsidless (+)ssRNA viruses, transfection with in vitro-synthesized full-length genomic RNA[15,16] or transformation with full-length cDNA[17,18] are available. Protoplast fusion-based virus horizontal transmission was shown to be useful between virus-infected donors and many recipients for any viruses, i.e., encapsidated and capsidless viruses[19,20].

*Neurospora crassa* (family Sordariaceae) is a filamentous ascomycete used as a research material for the one-gene–one-enzyme hypothesis by Beadle and Tatum in 1941[21]. This fungus has served as a model eukaryotic multicellular organism for genetics, developmental biology, and molecular biology. In particular, circadian rhythm-based physiological regulation[22], RNA interference (RNAi) post-transcriptional gene silencing[23,24], and DNA methylation-mediated epigenetic control[25] have been pioneered by researchers using *N. crassa*. The advantages of this fungus over other organisms, especially over other filamentous fungi, include the public availability of a number of biological and molecular tools, biological tractability[26], i.e., rapid vegetative growth and ease of sexual mating, shared techniques, a well-annotated genome sequence[27], and single-gene knockout (KO) collection[28] available from the Fungal Genetics Stock Center (FGSC) (http://www.fgsc.net)[29]. Surprisingly, despite these merits, this fungus is not utilized in virological studies.

RNAi, also known as RNA silencing, occurs in a wide variety of eukaryotic organisms, including animals and plants. The basic process of RNAi involves dsRNA that is recognized and diced by Dicer into small RNAs. These small RNAs then are incorporated into the effector Argonaute complex for repression of the target gene expression at transcriptional and post-transcriptional levels[30,31]. RNAi pathways in fungi were deciphered for the first time in *N. crassa*[23,24,32]. The known pathways in *N. crassa* operate at the the the post-transcriptional level, and are largely divided into two groups: meiotic and mitotic RNAi depending on which stage RNAi occurs in. The second group is further categorized into two: quelling and qiRNA (QDE-2-interacting small RNA)-mediated silencing depending on how dsRNA is generated[33,34]. *N. crassa* has three RNA-dependent RNA polymerase (RDR), two Dicer (Dicer-like DCL), and two Argonaute (Argonaute-like AGL) proteins. Two vegetative RNAi pathways share players, while aberrant RNA is generated from repetitive chromosomal regions and the UV-damaged nuclear genome, respectively. Although RNAi in some fungi has been shown to act as an antiviral defense-system response as in the case of animals and plants, its role in antiviral response has not been demonstrated in *N. crassa*. Choudhary et al. demonstrated the transcriptional induction of RNAi pathway and other putative antiviral genes by transgenic expression of dsRNA, suggesting that the RNAi pathway may act as part of the viral defense

mechanism in this fungus[35]. In this sense, a phytopathogenic ascomycete, *Cryphonectria parasitica* (chestnut blight fungus, family Cryphonectriaceae), has been used to dissect antiviral RNAi in the past decade. Of four RDRs, two Dicer, and four Argonaute genes, one Dicer (*dcl2*) and one Argonaute (*agl2*) gene play major roles in antiviral RNAi[36,37]. Of note is that the key genes are transcriptionally upregulated upon virus infection, for which a general transcriptional activator SAGA (Spt–Ada–Gcn5 acetyltransferase) complex and DCL2 (positive-feedback player) are essential[38,39].

Here, we demonstrate the detection of diverse RNA viruses from *N. crassa* and two other *Neurospora* species *N. intermedia* and *N. discreta*, and experimental introduction into the *N. crassa* standard strain of several viruses originally isolated from other fungi using three different methods. Utilizing the biological resources and molecular tools of *N. crassa* led to interesting insights into how virus/host interactions, particularly antiviral RNAi/viral counter-RNAi, are regulated. Collectively, this study establishes a foundation for the study of virology in the model organism *N. crassa*.

## Results

**Diverse RNA viruses in *Neurospora* spp**. To elucidate virus infection in wild *Neurospora* species, we first chose six wild *N. intermedia* isolates from Bogor, Indonesia (FGSC strains #2558, #2559, #5098, #5099, #5643, and #5644) and performed a small-scale screen by a conventional method. The presence of dsRNA of ~8 kbp and 5 kbp was observed in the FGSC #2559 and FGSC #5099, respectively (Fig. 1a). The complete sequence of a dsRNA segment from FGSC #2559 determined by next-generation and Sanger sequencing was shown to represent the replicative dsRNA form of a capsidless (+)ssRNA virus, termed Neurospora intermedia fusarivirus 1 (NiFV1, a putative member of the proposed family "Fusariviridae") (DDBJ/EMBL/GenBank accession #: LC530174). The dsRNA of ~5.0 kbp in FGSC #5099 has yet to be characterized.

Mining of virus sequences from transcriptomic data has been a trend for hunting and discoveries of viruses[9,40]. There are publicly available wealthy transcriptomic data of *N. crassa*[41,42], prompting us to mine for virus sequences infecting this fungus. In total, six different types of RNA viruses, specifically four (+) ssRNA viruses (e.g., fusarivirus, deltaflexivirus, alpha-like virus, and ourmia-like virus), one dsRNA virus (partitivirus), and one uncategorized RNA virus, were detectable in each data set for 15 out of 134 different *N. crassa* field-collected strains (Table 1 and Supplementary Table 1). To confirm the virus-like sequences in vivo, we detected dsRNAs from 12 out of 15 strains (Fig. 1b). We also obtained 16 well-assembled fragments of fusariviruses divided into three subgroups (namely A, B, and C in clade I) from 12 fungal strains of different geographical origins and found that 7 fungal strains were infected by a single fusarivirus and 5 other strains were infected by two fusariviruses (Table 1). These suggest a widespread nature of fusariviruses (see below) in members of the genus *Neurospora* of different geographical origins (Table 1, Fig. 2, and Supplementary Fig. 1). Among them, a single fusarivirus (subgroup A) harbored in the Louisiana isolate JW60, closest to the standard *N. crassa* strain 74-OR23-1VA, was fully determined by Sanger sequencing as the standard *N. crassa* fusarivirus (DDBJ/EMBL/GenBank accession #: LC530175), termed Neurospora crassa fusarivirus 1 (NcFV1) (Fig. 2a). NcFV1-JW60 was used in the subsequent investigation into virus/host interactions, because the standard *N. crassa* strain singly infected by this virus could readily be obtained. Similarly, we completely sequenced the genomic segments, dsRNA1 and dsRNA2, of a partitivirus termed Neurospora crassa partitivirus 1

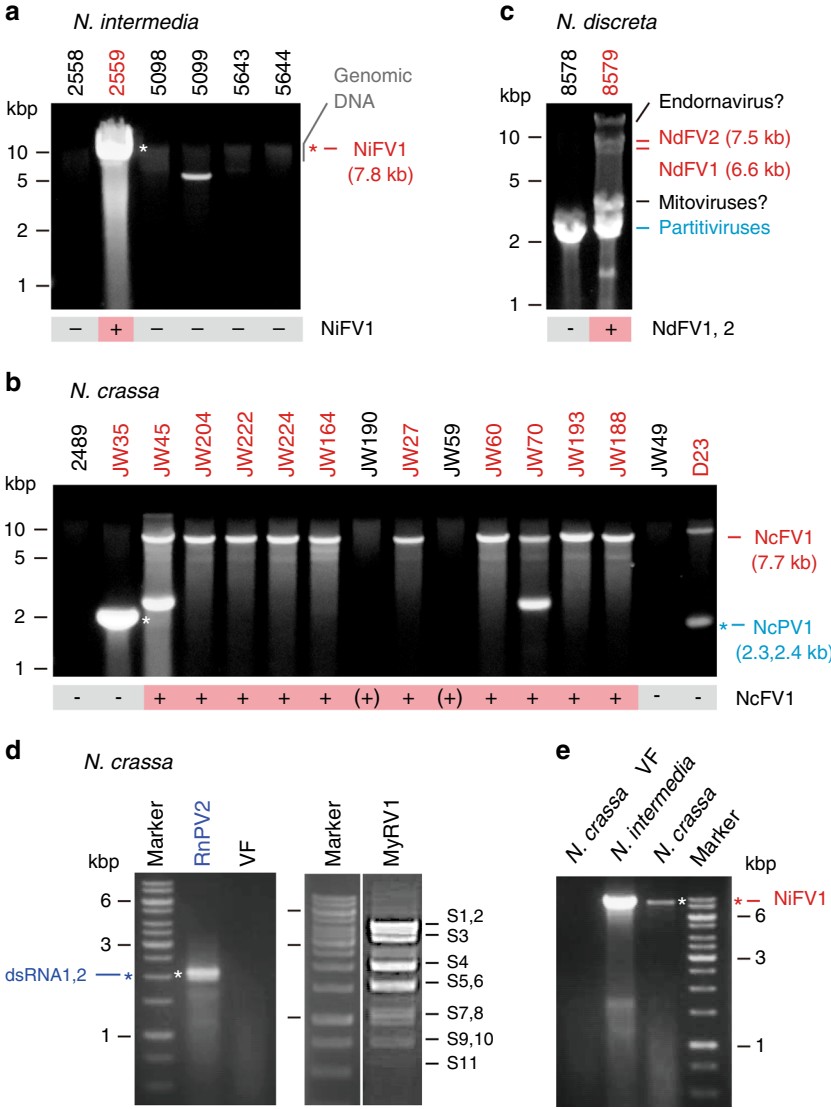

**Fig. 1 Virus detection in *Neurospora* spp. and experimental introduction of heterologous viruses into the *Neurospora crassa* strain.** Agarose gel electrophoresis of dsRNA fractions obtained from *N. intermedia* (**a**), *N. crassa* (**b**), and *N. discreta* (**c**). After electrophoresis, dsRNA extracted from isolates of *Neurospora* spp. shown on the top of each gel was stained with GelGreen (Biotium, Inc.). Fully sequenced viruses are Neurospora intermedia fusarivurs 1 (NiFV1), Neurospora crassa fusarivurs 1 (NcFV1), and Neurospora crassa partitivirus 1 (NcPV1). Multiple dsRNA bands were detected in *N. discreta* isolate FGSC #8549 that is expected to represent the genomes of several RNA viruses (Table 1). The 15 field-collected *N. crassa* strains were subjected to dsRNA analyses together with a potential virus-free standard strain FGSC #2489 (Supplementary Table 1). The discrete dsRNAs of ~2–3 kbp detectable in *N. crassa* JW45, JW70, and D23 may be defective RNAs. **d** Experimental introduction of encapsidated dsRNA viruses into *N. crassa* via virion transfection. Virions of Rosellinia necatrix partitivirus 2 (RnPV2) and mycoreovirus 1 (MyRV1) were purified from their original host fungi and transfected into protoplasts of the standard strain of *N. crassa*. DsRNA was isolated from the representative transfectant by RnPV2 and MyRV1 as well as the virus-free standard strain (VF) and electrophoresed in agarose gel. Lane M indicates 1-kb DNA ladder (Thermo Fisher Scientific). **e** Experimental introduction of a putative capsidless (+) ssRNA virus into *N. crassa*. Protoplasts of *N. intermedia* (FGSC #2559) infected by a fusarivirus NiFV1 and the standard *N. crassa* strain were fused. Each of the experiments shown in panels **a–e** was independently repeated at least twice.

(NcPV1) from the wild Floridan isolate JW35 (DDBJ/EMBL/GenBank accession #: LC530176 and LC530177) (Fig. 3a).

*Neurospora discreta* strain FGSC #8579 (Belen, New Mexico, USA), which is mostly used as a standard strain, has recently been suggested to carry a fusarivirus, Neurospora discreta fusarivirus 1 (NdFV1) using the publicly available transcriptomic data SRR1539773[9]. Similarly, we took an in silico approach and detected not only NdFV1 but also many other virus-like sequences: another fusarivirus and four partitiviruses, termed Neurospora discreta fusarivirus 2 (NdFV2), Neurospora discreta partitivirus 1–4 (NdPV1–4), and two mitoviruses (capsidless (+) ssRNA viruses) and an endornavirus [a capsidless (+)ssRNA

virus] (Table 1). Indeed, dsRNA gel electrophoresis revealed multiple bands in the strain (Fig. 1c).

Sequence and phylogenetic analyses of the characterized fusariviruses of *Neurospora* spp. showed that (1) viral RdRP was more conserved than the P2 protein (RdRP: 29.94–99.33%, P2: 14.96–99.48%), (2) NcFV1, NiFV1, and NdFV2 belong to the same clade (I) and share larger genome size, while NdFV1 contains a coiled-coil domain in P2 and belongs to another clade (II), and (3) *N. crassa* fusariviruses appeared to be separated into three subgroups, NcFV1A, B, and C in clade I, based on the phylogenetic relationship and amino acid sequence identity of RdRP and P2 (Fig. 2). All five *Neurospora* partitiviruses, NcPV1

**Table 1 List of *Neurospora* viruses and their host fungal strains.**

| FGSC # | Strain name | Geographic origin | Mating type | SRA file | Virus (virus abbreviation: accession no., subgroup type) |
|---|---|---|---|---|---|
| *Neurospora crassa* | | | | | |
| 2489 | 74-OR23-1VA | Marrero, Louisiana, U.S.A. | A | SRR797950 | Virus free (a standard, wild-type strain) |
| 3975 | JW35 | Florida, U.S.A. | a | SRR08983 | Partitivirus (NcPV1:LC530176, LC530177) |
| 4713 | JW45 | Merger, Haiti | A | SRR089840 | Fusarivirus (NcFV1:LC586023, B type), alpha-like virus/P3437 |
| 10651 | JW204 | Bayou Chicot, LA, U.S.A. | A | SRR798021 | Fusariviruses (NcFV1, A and B types) |
| 10658 | JW222 | Coon, LA, U.S.A. | a | SRR798029 | Fusarivirus (NcFV1:LC586028, C type) |
| 10659 | JW224 | Coon, LA, U.S.A. | a | SRR798030 | Fusarivirus (NcFV1, C type) |
| 10899 | JW164 | Marrero, LA, U.S.A. | a | SRR797998 | Fusariviruses (NcFV1:LC586025, A and B types), alpha-like virus |
| 10913 | JW190 | Elizabeth, LA, U.S.A. | A | SRR798014 | Fusariviruses (NcFV1:LC586026, A and B types) |
| 10948 | JW27 | Bayou Chicot, LA, U.S.A. | A | SRR798051 | Fusarivirus (NcFV1:LC586022, C type) |
| 10949 | JW59 | Coon, LA, U.S.A. | a | SRR798053 | Fusariviruses (NcFV1:LC586024, A and C types) |
| 10950 | JW60 | Coon, LA, U.S.A. | a | SRR798054 | Fusarivirus (NcFV1: LC530175, A type) |
| 10951 | JW70 | Coon, LA U.S.A. | A | SRR798057 | Fusariviruses (NcFV1, A and B types) |
| 10983 | JW193 | Elizabeth, LA, U.S.A. | a | SRR798015 | Fusarivirus (NcFV1:LC586027, B type) |
| 10912 | JW188 | Elizabeth, LA, U.S.A. | A | SRR798013 | Fusarivirus (NcFV1), ourmia-like virus |
| 4716 | JW49 | Puilboreau Mt., Haiti | A | SRR089843 | Ourmia-like virus |
| 8783 | D23 | Florida, U.S.A. | A | SRR089764 | Deltaflexivirus, alpha-like viruses |
| *Neurospora intermedia* | | | | | |
| 2558 | H2121 | Bogor Pasar, Indonesia | A | n.a. | Not detected |
| 2559 | HC2125-1 | Bogor Pasar, Indonesia | a | DRR248874 | Fusarivirus (NiFV1:LC530174) |
| 5098 | H2156 | Pasar Balubur, Indonesia | A | n.a. | Not detected |
| 5099 | H2158 | Pasar Balubur, Indonesia | a | n.a. | Unknown virus-like agent (~5.0 kbp) |
| 5643 | P0151 | Bogor, Indonesia | A | n.a. | Not detected |
| 5644 | P0153 | Bogor-4, Indonesia | A | n.a. | Not detected |
| *Neurospora discreta* | | | | | |
| 8579 | W683 | Belen, New Mexico, U.S.A. | A | SRR1539773 | Fusariviruses (NdFV1, 2), partitiviruses (NdPV1–4), mitoviruses, and endornavirus |
| 8578 | W682 | Belen, New Mexico, U.S.A. | a | n.a. | Partitiviruses |

*NcFV1* Neurospora crassa fusarivirus 1, *NiFV1* Neurospora intermedia fusarivirus 1, *NdFV1, 2* Neurospora discreta fusarivirus 1, 2 (LC586029, LC586030), *NcPV1* Neurospora crassa partitivirus 1, *NdPV1–4* Neurospora discreta partitivirus 1–4 (LC586031-LC586038), *n.a.* not available.

and NdPV1–4, share molecular attributes with other known partitiviruses; dsRNA1 encodes RdRP and dsRNA2 encodes CP (Fig. 3a, b). They have all been shown to belong to the genus *Betapartitivirus* unlike the alphapartitivirus RnPV2 (genus *Alphapartitivirus*) able to replicate in *N. crassa*[43] (Fig. 3c). Of note is that NdPV1 shares over 97% CP and RdRP amino acid sequence identity with *N. crassa* partitivirus NcPV1. This fact suggests that these two viruses belong to the same viral species in the *Betapartitivirus*, and that interspecies virus transmission may have occurred between *N. crassa* and *N. descreta*.

**Experimental introduction of heterologous viruses into *N. crassa*.** To elucidate whether *N. crassa* is able to support the replication of heterologous viruses originated from other fungal families, we chose two relatively well-characterized dsRNA viruses: Rosellinia necatrix partitivirus 2 (RnPV2, genus *Alphapartitivirus*, family *Partitiviridae*)[44,45] isolated from the white root rot fungus (*Rosellinia necatrix*, family Xylariaceae) and mycoreovirus 1 (MyRV1, genus *Mycoreovirus*, family *Reoviridae*)[11] isolated from the chestnut blight fungus (*C. parasitica*). Using purified preparation, the two viruses were transfected into spheroplasts of the standard strain of *N. crassa*. Figure 1d shows the dsRNA profiles of representative colonies infected by the respective virus. Specific genomic dsRNA bands were observed: dsRNA1 and dsRNA2 for RnPV2, and S1–S11 for MyRV1.

An interesting difference was noted; the ratio of dsRNA1: dsRNA2 was smaller in the natural *R. necatrix* host than in the newly expanded *N. crassa* host. This is not so surprising for multisegmented, multiparticulate viruses such as partitiviruses in which each genomic segment is packaged separately[44,46]. This is different from reoviruses that are multisegmented, monoparticulate viruses in which a set of the multiple genomic segments are packaged in a single particle. Indeed, similar phenomena were observed earlier in natural and experimental host fungi after virion transfection[44,47]. In contrast, the MyRV1 pattern was indistinguishable from the original infection of *C. parasitica*[48]. It should be noted that, unlike RnPV2, MyRV1 showed unstable infection during the subculture of transfected colonies.

For capsidless viruses, we previously established protoplast fusion techniques for virus introduction[20], because virion transfection was not applicable. This method was tested for the putative capsidless RNA virus NiFV1 (a fusarivirus) originally hosted by *N. intermedia* (Fig. 1a). Protoplasts were prepared from the donor strain FGSC #2559 of *N. intermedia* and the recipient strain of *N. crassa* (the standard strain 74-OR23-1VA) that harbored a nourseothricin (NTC)-resistance gene. After protoplast fusion, recipient colonies were selected on potato-dextrose agar (PDA) supplemented with NTC (PDA–NTC). Several of them tested positive for NiFV1 and repeatedly anastomosed with the virus-free, NAT-resistant strain of *N. crassa*. The genetic background of NiFV1-infected recipients was shown to be *N.*

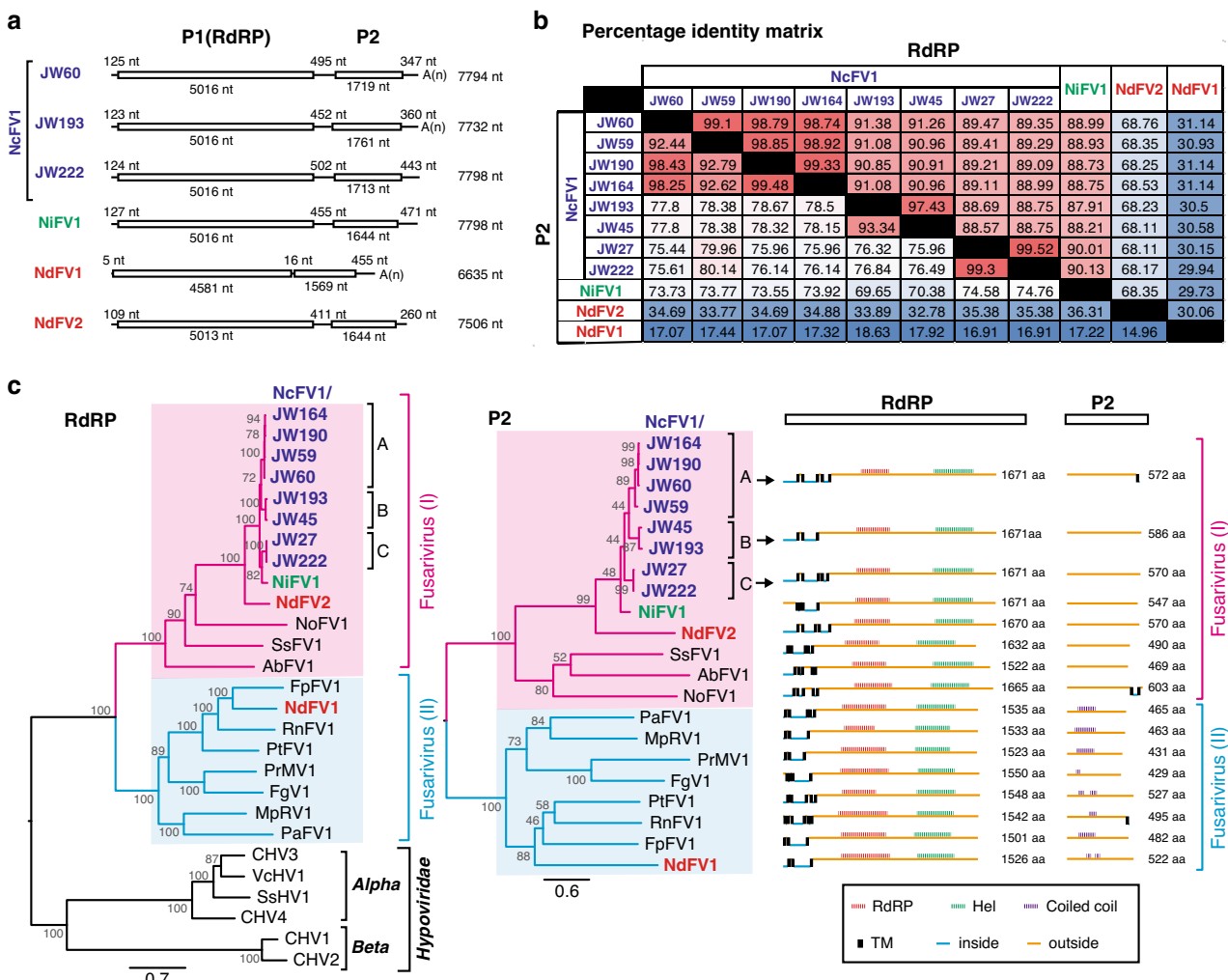

**Fig. 2 Sequence and phylogenetic analyses of fusariviruses detected from the strains of *Neurospora* species. a** Genome structure of Neurospora crassa fusarivirus 1 (NcFV1), Neurospora intermedia fusarivirus 1 (NiFV1), and Neurospora discreta fusarivirus 1 and 2 (NdFV1 and 2). The open boxes indicate open-reading frames (ORFs) with their lengths shown below. The lengths of the 5′- and 3′-untranslated regions, and intergenic regions, are indicated in nucleotides. **b** Percent identity matrix (PMI) among *Neurospora* fusariviruses generated by Clustal-Omega <https://www.ebi.ac.uk/Tools/msa/clustalo/>. The top half is the PMI of the viral RNA-dependent RNA polymerase (RdRP) and the bottom half is the PMI of ORF2 protein (P2, unknown function). **c** The maximum-likelihood (ML) phylogenetic trees were constructed based on alignment of the entire region of RdRP and P2 using RAxML-NG with 1000 bootstrap replications (left and middle panels). A LG+I+G4+F model was selected as a best-fit model for the alignment using ModelTest-NG. The trees are visualized by FigTree with the midpoint-rooting method. The two subclades of *Neurospora* fusariviruses (I and II) and three subgroups of *N. crassa* fusarivirus NcFV1 (A, B, and C) are indicated. The selected members of the family *Hypoviridae* are included as outgroups. The numbers on the left of branches indicate the percent bootstrap values. Putative secondary structures and functional domains of the RdRP and P2 proteins are shown on the right. Predictions of conserved domains and inside or outside of the membrane are predicted by SMART and TMHMM, respectively. Hel and TM indicate predicted helicase and transmembrane domains, respectively. Virus abbreviations are summarized in Supplementary Table 4.

*crassa* by the observation that the recipient could anastomose with the original isogenic recipient strain. Interestingly, NiFV1 dsRNA-replicative form accumulated much less in *N. crassa* than in the original host, *N. intermedia* when normalized to starting mycelia (Fig. 1e). NiFV1 appears not to be adjusted well to *N. crassa*. Similar phenomena were observed when virus contents were compared between their original and experimental host fungi[13,44,49].

In summary, *Neurospora* spp. was shown to host a variety of viruses that belong to at least six RNA virus families and an unclassified RNA virus group: *Fusarivirdae*, *Partitiviridae*, *Endornaviridae*, *Reoviridae*, *Deltaflexiviridae*, *Narnaviridae*, and an unclassified RNA virus group (alpha-like viruses). Among these, fusariviruses appeared to prevail in *Neurospora* spp. (Table 1 and Supplementary Fig. 1).

**Two Dicers and one Argonaut play a major role in antiviral defense during the vegetative phase in *N. crassa*.** Among *Neurospora* viruses, fully sequenced NcFV1 and NcPV1 were horizontally transferred between vegetatively compatible and incompatible strains of *N. crassa*. The two viruses were first moved to the *N. crassa helper-5* strain (FGSC #8747; Δ*mat his-3 tk+ hph cyh-1, Bml pan-2*) from the original host strains, NcFV1-infected FGSC #10950 (JW60), and NcPV1-infected FGSC #3975 (JW35), by coculture. The *helper-5* strain has been used for forcing and resolving heterokaryons (Supplementary Fig. 2). The heterologous virus RnPV2 but not MyRV1 was stably transferred into the *helper-5* strain from the transfected *Neurospora* strains. After confirmed to be virus-infected, the recipient *helper-5* strains were used as donors to transmit NcFV1, NcPV1, and RnPV2 to the standard *N. crassa* strain (74-OR23-1VA) and its derivatives.

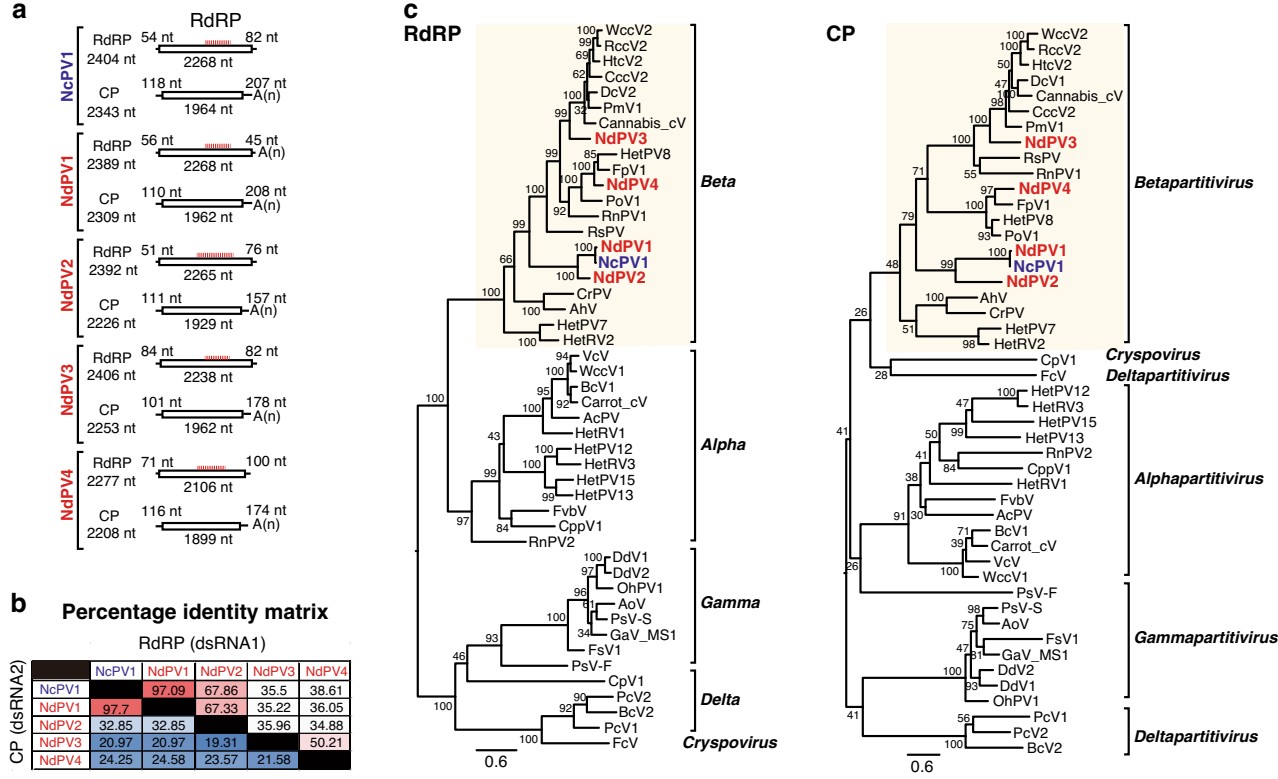

**Fig. 3 Sequence and phylogenetic analyses of partitiviruses detected from the strains of *Neurospora* species. a** Genome structure of Neurospora crassa partitivirus 1 (NcPV1), and Neurospora discreta partitivirus 1–4 (NdPV1–4). The clear boxes indicate open-reading frames (ORFs) with their lengths shown below. The lengths of the 5′- and 3′-untranslated regions, and intergenic regions are indicated in nucleotides. **b** Percent identity matrix (PMI) among partitiviruses detected from two *Neurospora* species, *N. crassa*, and *N. discreta*. See Fig. 1b legend for detailed explanation. **c** The maximum-likelihood (ML) phylogenetic trees based on alignment of the entire region of RdRP and CP using RAxML-NG. See Fig. 1c legend for methodology. Virus abbreviations are summarized in Supplementary Table 4.

Hereafter, these three viruses were largely utilized in subsequent deeper analyses of virus/host interactions.

We tested for enhancement of NcFV1, NcPV1, and RnPV2 accumulation a total of 14 single- and two double- and one-triple deletion mutant of the standard *N. crassa* strain with deletions of genes related to two RNAi pathways, mitotic silencing (quelling: *qde-1*, *qde-2*, *qde-3*, *dcl-1*, *dcl-2*, *qip*, and *rpa-3*) in the vegetative stage and meiotic silencing by unpaired DNA (MSUD: *Sad-1*, *Sad-2*, *sad-3*, *Sad-4*, and *sad-5*, *dcl-1*, *qip*, and *sms-2*)[34], or other function (Supplementary Table 2). To this end, we quantitatively compared viral dsRNA and ssRNA by agarose gel electrophoresis of dsRNA and RT-qPCR of total RNA fractions, respectively, and found that they exhibited similar accumulation profiles in the *N. crassa* mutant strains (Fig. 4a and Supplementary Fig. 3). Of the single-deletion mutants tested, only Δ*qde-2* showed ~10-fold and ~25-fold elevation in accumulation of NcFV1 dsRNA-replicative form and viral ssRNA, relative to the wild-type strain, respectively (Fig. 4a). The two dicer genes were anticipated to work redundantly in antiviral defense as in the case for transgene silencing (quelling)[50]. To test this hypothesis, a double *dcl* mutant (Δ*dcl-1/2*) was prepared and infected by NcFV1. Consequently, the double mutant (Δ*dcl-1/2*) showed much greater susceptibility to NcFV1 than the single *dcl* mutants or wild type, indicating the redundant function of the two Dicers against this virus (Fig. 4a). To examine similar redundancy in Argonaute and RDR, we created their double- (Δ*qde-2*/Δ*sms-2*) and triple (Δ*qde-1*/Δ*Sad-1*/Δ*rrp-3*) deletion mutants, respectively. No or slightly elevated (~1.9-fold) accumulation of NcFV1 dsRNA and ssRNA was observed in the double Argonaute mutant (Fig. 4a, Δ*qde-2* and Δ*Sms-2*) compared to the *qde-2* single mutant (Δ*qde-2*) showing

that QDE-2, but not SMS-2, predominantly functions in the antivirus RNAi against this virus. Similarly, NcFV1 accumulated in a triple *rdr* mutant (Δ*qde-1*/Δ*Sad-1*/Δ*rrp-3*) at a level similar to that in their single-deletion mutants, suggesting no involvement of these *rdr* genes in the antiviral RNAi.

By contrast, Δ*rrp-3*, Δ*dcl-2*, and Δ*qde-2* showed only ~1.4-fold or <3.7-fold increased accumulation levels of NcPV1 genomic dsRNA or its transcripts compared with any other single-deletion mutants (Fig. 4b and Supplementary Fig. 3b). No significant additional increase in NcPV1 accumulation, relative to Δ*rrp-3*, Δ*dcl-2*, and Δ*qde-2*, was observed in the Dicer double and RDR triple mutants. The Argonaute double mutant Δ*qde-2*/Δ*Sms-2*, manifested two-to-threefold increased transcript levels relative to the single mutants (Fig. 4b). Similarly, no over twofold change in RnPV2 genomic dsRNA accumulation in the tested mutants from that in the wild-type strain was observed (Fig. 4c and Supplementary Fig. 3c). When compared with the wild-type strain, no >1.7-fold increased accumulation of RnPV2 transcripts was detected, whereas a two-to-threefold decrease in RnPV2 transcript level was observed in Δ*qde-2*, Δ*dcl-1/2*, and Δ*qde-2*/Δ*Sms-2* (Fig. 4c), a phenomenon warranting further investigation.

Collectively, we demonstrated that two Dicers DCL-1 and DCL-2 and one Argonaut QDE-2 play a major role in antiviral RNAi against at least a fusarivirus in the vegetative phase of *N. crassa*.

**Transcriptional and post-transcriptional regulation of host genes upon virus infection.** Many genes including those of the RNAi pathway and putative antiviral response were previously

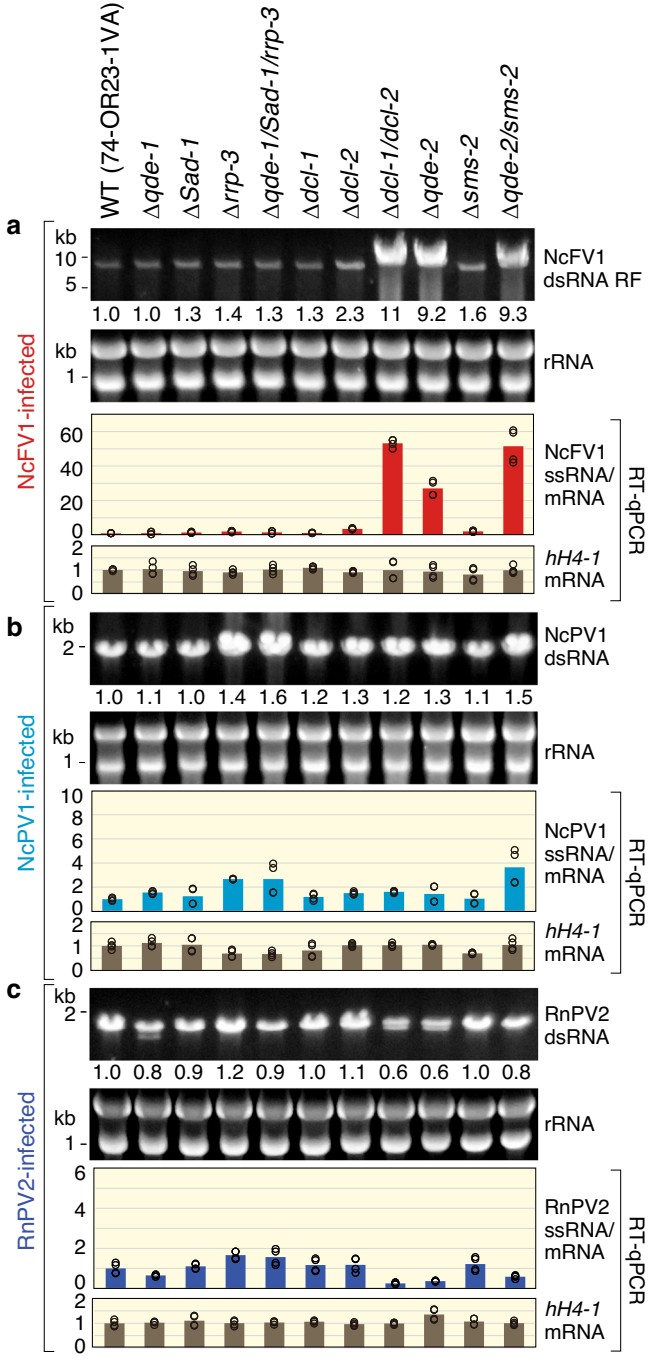

**Fig. 4 Quantitative comparison of virus accumulation in different mutants of *Neurospora crassa*.** Accumulation of Neurospora crassa fusarivirus 1 (NcFV1) (**a**), Neurospora crassa partitivirus 1 (NcPV1) (**b**), and Rosellinia necatrix partitivirus 2 (RnPV2) (**c**) was compared among *N. crassa* mutants with deletion of RNAi-related genes shown on the top of each gel. The mutants have the same genetic background as the standard *N. crassa* strain, 74-OR23-1VA. Total RNA fractions were extracted from different mutants as well as the wild-type 74-OR23-1VA strain. After normalization against ribosomal RNA (rRNA) as shown on the middle panels, the obtained dsRNA fractions were electrophoresed and stained with GelGreen (Biotium, Inc.) as for Fig. 1. Viral dsRNA or dsRNA-replicative form (RF) accumulation values (means) were obtained from two biological replicates by ImageJ, and are shown below each dsRNA gel, where virus accumulation in the wild-type strain was expressed as 1. Accumulation of positive-sense (+) single-stranded (ss) RNA of the three viruses was also compared among the aforementioned *N. crassa* mutants by real-time RT-qPCR. Total RNA fractions were obtained from different mutants as well as the wild-type standard strain. Beta-tubulin mRNA (NCU04054) was used as an internal control. Histone H4 (hH4) mRNA was also monitored. RT-qPCR was performed as described in "Methods" using the primers shown in Supplementary Table 3. Bars denote mean values calculated from a total of four replicates (two biological and two technical replicates), while open circles indicate obtained relative accumulation values for single replicates.

the aforementioned virus hunting (see Supplementary Table 1). Some of the other *N. crassa* genes identified as dsRNA-inducible genes by Choudhary et al[35]. were also upregulated upon virus infection, which included NCU04490 (6–16 family), NCU07036 (3′–5′ exonuclease), NCU04472 (RNA helicase), NCU09495 (*set-6*), and NCU00947 (unknown function) (Supplementary Fig. 5).

To elucidate the transcriptional activation of RNAi-related genes upon virus infection, we performed chromatin immuno-precipitation (ChIP) followed by RT-qPCR. Dimethylation of histone H3 Lys 4 (H3K4me2) and Ser 5 phosphorylation of the RNA polymerase II C-terminal domain (Pol II S5P-CTD) are well-characterized as transcriptionally activated markers. As expected, significant and moderate accumulation of H3K4me2 and Pol II S5P-CTD was observed at the *dcl-2* and *qde-2* loci upon NcFV1, NcPV1, or RnPV2 infection, respectively (Fig. 5c). However, the *rrp-3* gene locus and the noninducible genes *dcl-1* and *qde-1* loci did not appear to be enriched by H3K4me2 and Pol II S5P-CTD, when NcFV1 or NcPV1 was present (Fig. 5c). RnPV2 led to modest accumulation of Pol II S5P-CTD, but not H3K4me2, in the *rrp-3* gene locus and the noninducible gene *dcl-1* locus (Fig. 5c). Alternative transcriptional activation or post-transcriptional RNA processing might be involved in the increase of the *rrp-3* mRNA.

The results prompted us to examine their protein accumulation levels upon virus infection. We created strains expressing FLAG-octapeptide-tagged DCL-2, QDE-2, or RRP-3 from their endogenous loci and infected them by NcFV1, NcPV1, and RnPV2. Western blotting revealed that DCL-2 and RRP-3 were strikingly elevated upon virus infection (Fig. 5d). Surprisingly, QDE-2 was shown to accumulate much less in the strain infected by NcFV1, but not in that by NcPV1 or RnPV2, than in the virus-free standard strain (Fig. 5d), suggesting NcFV1-specific post-transcriptional downregulation of *qde-2*. This downregulation may result from a counterdefense response targeting QDE-2 directly or indirectly by NcFV1.

Taken together, these combined results clearly show transcriptional and post-transcriptional regulation of key genes of the antiviral RNAi in *N. crassa* upon virus infection.

reported to be induced at the transcription level by transgenic dsRNA expression in *N. crassa*[35]. In another filamentous asco-mycetous fungus, *C. parasitica*, infection by some RNA viruses was shown to induce similar homologous genes[37,51]. Thus, it was of interest to investigate whether virus infection triggers tran-scriptional induction of the *N. crassa* genes known to be induced by dsRNA. As shown in Fig. 5a, transcription of *dcl-2* and *rrp-3* was strikingly upregulated in the standard *N. crassa* strain fol-lowing infection by NcFV1, NcPV1, or RnPV2, while moderate transcriptional upregulation of *qde-2* was confirmed following infection by NcFV1, NcPV1, or RnPV2 (Fig. 5b). No or a few transcriptional changes were observed in *qde-1* and *dcl-1* upon virus infection. Furthermore, a similar result was obtained by differential gene expression (DGE) analysis using the available *N. crassa* RNA-seq data (Supplementary Fig. 4), which were used for

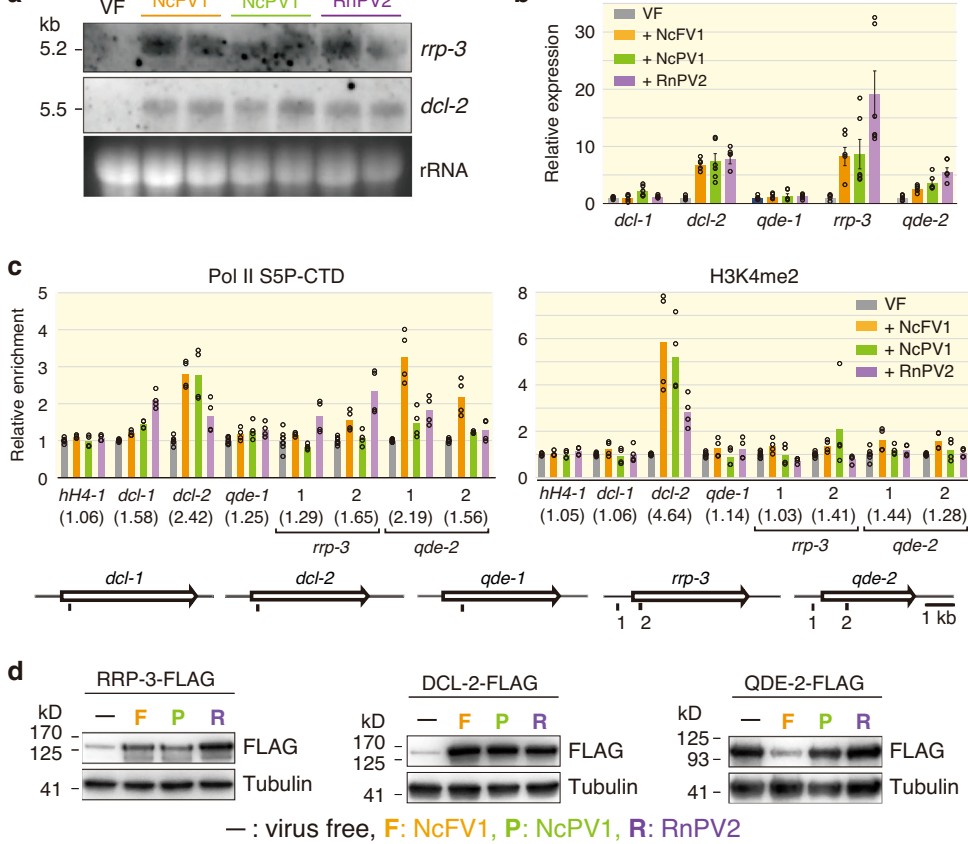

**Fig. 5 Transcriptional and post-transcriptional regulation of RNAi-related genes in *Neurospora crassa*. a** Northern blotting of *rrp-3* and *dcl-2* in NcFV1-, NcPV1-, or RnPV2-infected standard *N. crassa* strains, 74-OR23-1VA strains. **b** Real-time RT-qPCR analyses of several RNAi-related genes. Transcript levels of *dcl-1*, *dcl-2*, *qde-1*, and *qde-2* were compared between a virus-free (VF) standard *N. crassa* strain (74-OR23-1VA) and its virus-infected derivative strains by NcFV1, NcPV1, and RnPV2. Histone H4 mRNA (*hH4-1*) was used as an internal control in (**b**) and (**c**). Mean values and standard deviations were calculated from three biological replicates and three technical replicates. A total of six obtained values are placed on each bar. **c** Chromatin immunoprecipitation (ChIP) assay of the standard *N. crassa* strain uninfected (VF) or infected by NcFV1, NcPV1, or RnPV2. DNA was isolated from immune precipitates with anti-H3K4me2 antibody (H3K4me2) or anti-RNA polymerase II CTD repeat YSPTSPS (Pol II S5P-CTD). Positions 1 and 2 refer to the amplified regions of the tested genes. Mean values were calculated from two biological replicates and two technical replicates. A total of four obtained values are placed on each bar. **d** Western blotting of RRP-3, DCL-2, and QDE-2. The standard *N. crassa* strain was engineered such that *rrp-3-Flag*, *dcl-2-Flag*, or *qde-2-Flag* was knocked in. Total protein fractions were prepared from the respective strain uninfected (−) or infected by NcFV1 (F), NcPV1 (P), or RnPV2 (R), were probed by anti-FLAG antibody. Alpha-tubulin was used a loading control. The experiments shown in panels **a** and **d** were repeated once and twice, respectively.

## Discussion

This study represents the establishment of *N. crassa* as a virus model host for studying virus/host interactions or virology in this well-studied model organism, based on three achievements. First, we report the discovery of diverse RNA viruses from different isolates of *N. crassa* and other *Neurospora* spp. (Table 1). *N. crassa* was also shown to naturally and experimentally host diverse viruses with different genome types (Table 1). Second, we developed methods for inoculation, which is often difficult for fungal viruses, of several capsidless and encapsidated RNA viruses in *N. crassa*. The natural horizontal transfer between fungal strains is generally hampered by a self-/non-self-recognition system operating at the intraspecies level[52–54]. We tested a few methods for virus introduction into *N. crassa* with the standard genetic background[27]: hyphal fusion (supplementary Fig. 2), protoplast fusion (Fig. 1e), and virion transfection (Fig. 1d)[11,14,19,20]. Third, utilizing available biological tools and molecular techniques, some in *N. crassa* genes restricting virus replication were identified and shown to be transcriptionally and post-transcriptionally regulated.

Fusariviruses, a group of potential capsidless (+)ssRNA viruses that are distantly related to hypoviruses (family *Hypoviridae*) and proposed as the family "Fusariviridae", were detected in different *Neurospora* spp. of different geographical origins such as the United States (Louisiana), Haiti, and Indonesia, suggesting its widespread nature in members of the genus *Neurospora* (Fig. 2). Only a few fusariviruses that have been molecularly and biologically investigated, among which are Fusarium graminearum virus 1 (FgV1) strain DK21[55–57] and Rosellinia necatrix fusarivirus 1 strain NW10 (RnFV1)[58]. These two fusariviruses appear to differ in genome organization and gene expression strategy. FgV1 has four open-reading frames (ORFs) and the three downstream ORFs are expressed via subgenomic RNAs, while RnFV1 has only two ORFs and the downstream ORF is not likely to be expressed via subgenomic RNAs but via an unknown mechanism. All fusariviruses identified from *Neurospora* spp. resemble RnFV1 in terms of the 2-ORF genome organization. Another difference from FgV1 was detected in host factor requirement. A *Fusarium graminearum* (family Nectriaceae) gene product, hexagonal peroxisome (Hex1) protein, was identified as

necessary for efficient FgV1 replication and normal symptom induction. Hex1 is a major component of the Woronin body, a peroxisome-derived organelle that fills the septal pore under hyphal wounding stress and prevents the extension of wound-induced damage to neighboring cells[59,60]. In this study, we examined the possible effects of deletion of the orthologous *hex-1* gene of *N. crassa* on NcFV1 replication. However, no discernable effect was observed (Supplementary Fig. 3), suggesting that the involvement of HEX1 in the fusarivirus replication is specific to the pathosystem: *F. graminearum*/FgV1.

*N. crassa* is the first organism to be used for genetic dissection of the RNAi pathway in fungi[23,24]. There are two types of RNAi: quelling and MSUD (meiotic silencing by unpaired DNA). Quelling corresponds to cytoplasmic mitotic transgene RNAi in other eukaryotes[61,62]. *N. crassa* has two Dicer (*dcl-1* and *dcl-2*), three RDR (*qde-1*, *Sad-1*, and *rrp-3*) and two Argonaute genes (*qde-2*, *sms-2*)[34,63]. Transgene-induced RNAi in *N. crassa* requires two Dicers (*dcl-1*, *dcl-2*), one RDR (*qde-1*), one Argonaute (*qde-2*, the homolog of *C. parasitica agl2* and *Caenorhabditis elegans rde1*), and RecQ DNA helicase (*qde-3*)[23,24,34,50]. However, what genes are required for antiviral RNAi in *N. crassa* was unknown. We have shown an essential role of *qde-2*-encoded Argonaute (QDE-2) in defense against NcFV1. This study clearly indicated the redundant functional role of DCL-1 and DCL-2 in antiviral RNAi as reported for quelling[50], that is, enhanced virus replication was only observed in the double-deletion mutant of *dcl-1* and *dcl-2*, but not in the respective single-deletion mutants (Fig. 4). These observations highlight differences from quelling in *N. crassa* and from antiviral RNAi in a model host filamentous fungus, *C. parasitica*[64,65] and similarities to two other ascomycetes, *Sclerotinia sclerotiorum* (family Sclerotiniaceae)[66] and *F. graminearum*[57] (Fig. 6). We demonstrated that all three RDRs (QDE-1, RRP-3, and SAD-1) were dispensable for antiviral RNAi in *N. crassa*, even though QDE-1 and SAD-1 play important roles in quelling and in MSUD. The dispensability of RDRs is reminiscent of antiviral RNAi in *C. parasitica* in which only *dcl2* and *agl2* are the two key genes[36,37]. While in *S. sclerotiorum*, *dcl-1*, *dcl-2*, and *agl-2* function in antiviral RNAi[66,67], in another ascomycete *F. graminearum*, two Dicers (FgDICER1 and FgDICER2) and two Argonautes (FgAGO1 and FgAGO2) function redundantly in antiviral RNAi[57,68]. Note that FgAGO1 is homologous to *N. crassa qde-2* and *C. parasitica agl2*.

In *N. crassa*, dsRNA induces transcriptional elevation of RNAi-related genes such as *dcl-2*, *qde-2*, and *rrp-3*[35]. Taking three approaches, we confirmed that these genes are also induced upon virus infection (Fig. 5a, b), as hypothesized by Choudhary et al.[35]. Similarly, Nuss and colleagues showed transcriptional induction of two key RNAi genes, *dcl2* and *agl2*, by dsRNA expression and virus infection in *C. parasitica*[37,51]. This transcriptional regulation requires DCL2 and SAGA (a universal transcriptional coactivator)[38,39]. Comparison of RNAi regulation between the two fungi reveals interesting conservation and differences. It is likely that dsRNA, regardless of viral or host origin, can trigger transcriptional induction, but not its small RNAs, suggesting that the dicing activity of Dicers is not required for the induction[39]. In contrast, an interesting difference was observed in the degree of induction: *dcl2* of *C. parasitica* was induced more highly than that in *N. crassa*, i.e., ~40-fold vs. ~8-fold, whereas the induction of *rrp-3* and its ortholog *rdr4* are comparably induced in the two fungi, i.e., 20–30-fold. However, *C. parasitica* RDR4 seems to be not fully functional due to a nonsense mutation in any of the three alternatively splicing variants of transcripts[69]. Thus, the biological significance of high transcriptional induction of *rdr4* in *C. parasitica* or *rrp-3* in *N. crassa* remains elusive. In *C. parasitica*, no redundancy was found in Dicer in antiviral RNAi, and *dcl2* and *agl2* transcript levels were increased 10- to 40-fold upon virus infection[37,70]. By contrast, two Dicer genes, *dcl-1* and *dcl-2*, in *N.*

*crassa* played redundant roles (Fig. 4a), and their transcript levels were augmented by less than 7-fold (Fig. 5b and Supplementary Fig. 4). The high transcriptional upregulation of *dcl-2* might have been compromised by the redundancy of Dicer during the course of evolution of filamentous fungi.

Different patterns between the accumulation of the partiti-viruses (NcPV1 and RnPV2) and NcFV1 in an array of mutant *N. crassa* strains were observed. In the double *dcl* mutant or *qde-2* mutants, NcFV1 dsRNA-replicative form accumulated approximately 10-fold relative to the wild-type strain, and this increase was more pronounced when NcFV1 ssRNA was compared (Fig. 4, Supplementary Fig. 3). Such an elevation in the two mutants was not observed for NcPV1 or RnPV2 dsRNA accumulation (Fig. 4, Supplementary Fig. 3). It was previously shown that certain partitiviruses are tolerant to antiviral RNAi, despite their ability to induce RNAi, and accumulate at a similar level in RNAi-competent and -deficient *C. parasitica* strains[47]. A similar phenomenon was also observed for a capsidless (+)ssRNA hypovirus[71]. Thus, the failure of NcPV1 and RnPV2 to accumulate more in *dcl-1/2* and *qde-2* mutants than in the wild-type strain is not surprising and suggests their evolution of ways to evade antiviral RNAi.

Collectively, this study has opened up an avenue in modern virology, and shall accelerate its advance with available molecular tools and biological resources. This has great impact on studies with viruses of other fungi, particularly plant pathogenic asco-mycetous fungi that share many homologous genes with *N. crassa*[72,73]. Some plant fungal diseases such as chestnut blight are targets of biological control using viruses infecting the pathogenic fungi so-called "virocontrol"[3,74,75]. Filamentous fungi have mul-tilayered antiviral defense impairing virocontrol: RNAi working at the cellular level and vegetative incompatibility functioning at the population level[53,65]. Better understanding of antiviral defense vs. viral counterdefense and fine-tuning of expression of asso-ciated genes are prerequisite for their successful virocontrol. In this regard, further studies using the *N. crassa*/viruses, e.g., aiming at exploring antivegetative incompatibility responses evoked by viruses[76], should contribute to virocontrol of phytopathogenic fungi.

We discovered post-transcriptional downregulation of an Argonaute QDE-2 specifically upon infection by a (+)ssRNA virus (NcFV1) in *N. crassa* (Fig. 6). A plant (+)ssRNA virus is known to encode an RNAi suppressor that induces an autophagy pathway targeting an Argonaute (AGO1)[77]. It will be of interest to explore the mechanism of NcFV1-mediated post-transcrip-tional downregulation of *qde-2* (Fig. 5d). It is anticipated that NcFV1 encodes an RNAi suppressor whose mode of action is different from those of the suppressors from the prototype hypovirus (Cryphonectria hypovirus 1 p29) and a fusarivirus (FgV1 ORF2 protein) that transcriptionally downregulate RNAi key genes[37,68,70]. Other interesting future challenges include to investigate whether meiotic silencing serves as antiviral defense, how the virus or dsRNA is sensed and triggers antiviral RNAi, how the virus impedes host fungal vegetative incompatibility, and what host factors are associated with viral replication and symptom induction.

## Methods

**Fungal and viral materials**. The fungal strains tested in this study are summarized in Table 1 and Supplementary Table 2. Many *N. crassa* strains were purchased from the Fungal Genetics Stock Center (FGSC) (http://www.fgsc.net) and *Neurospora* protocols available in the website were used unless otherwise mentioned. Full names and accession numbers of mycoviruses detected in this and previous studies are provided in Table 1. *R. necatrix* strains W57 (infected by RnPV2) and *C. parasitica* strain 9B21 (infected by MyRV1) were described earlier[11,44,78]. These fungal strains were grown on Difco^TM PDA plates for maintenance unless other-wise mentioned.

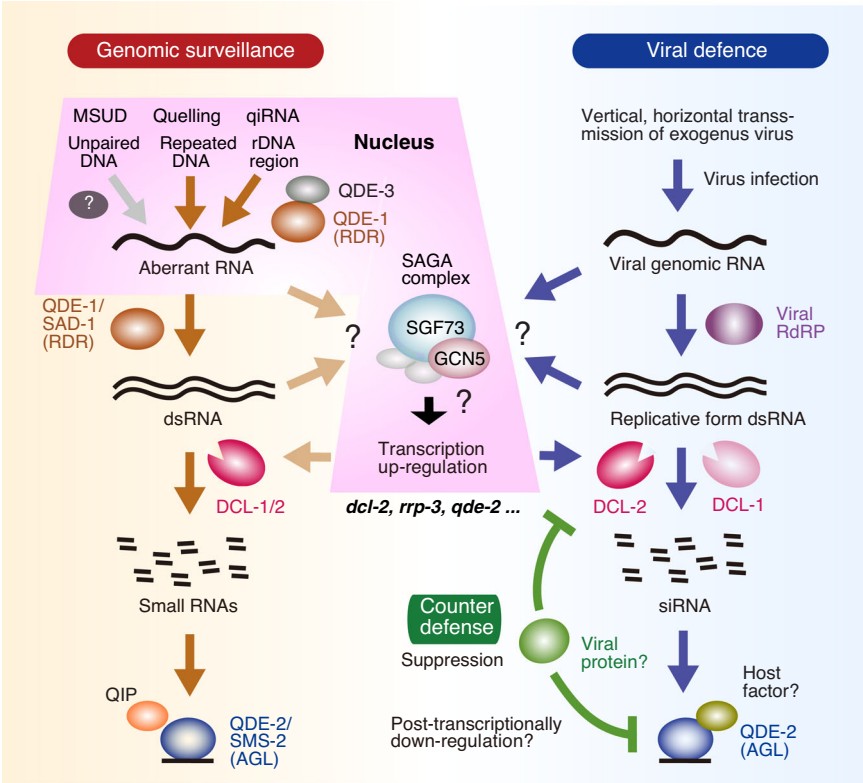

**Fig. 6 Model of antiviral RNAi and overall effects of virus infection on *Neurospora crassa*.** Proposed model of the antiviral RNAi pathway in parallel to the established transgene vegetative mitotic (quelling and qiRNA) and meiotic silencing by unpaired DNA (MSUD) pathways for genomic surveillance. The meiotic and mitotic RNAi pathway was modified from Chang et al.[34]. The major players in RNAi, RNA-dependent RNA polymerase (RdRP), or RDR (QDE-1, SAD-1), Dicer (DCL-1, DCL-2), and Argonaute (SMS-2, QDE-2), are shown in the pathways. Key players, DCL-1, DCL-2, and QDE-2 are shared by the antiviral and meiotic/mitotic pathways. As a counterdefense, viruses such as NcFV1 appear to deploy RNAi suppressors that post-transcriptionally downregulate RNAi key genes (see Fig. 5d). It remains unknown whether antiviral RNAi genes of *N. crassa* are transcriptionally repressed by viruses or whether the SAGA (*S*pt–*A*da–*G*cn5 acetyltransferase) complex and Dicer transcriptionally upregulate many *N. crassa* genes, as in the case for other filamentous ascomycetes, such as *Cryphonectria parasitica*.

*Neurospora* knock-in (KI) and KO strains (Supplementary Table 2) were prepared by the standard method as described earlier[28,79] with specific primers summarized in Supplementary Table 3.

**Experimental virus introduction into *N. crassa*.** Virion transfection was performed as described by Hillman et al.[11]. First, virus particles of RnPV2 and MyRV1 were prepared as described by Chiba et al.[44] and Hillman et al.[11]. Protoplasts of the *N. crassa* standard strain (74-OR23-1VA, FGSC #2489) were prepared by the method of Eusebio-Cope et al.[48] as it is generally applicable to protoplast preparation of ascomycetous fungi. Briefly, liquid cultures of *N. crassa* were harvested and incubated in a cell-wall digestion solution containing β-glucuronidase and lysing enzyme (Sigma-Aldrich). Protoplasts ($0.5 \times 10^7$ cells in 200 μl) were mixed with purified virus particles in the presence of polyethylene glycol (PEG) and $Ca^{2+}$ ion.

Protoplast fusion was pursued between *N. intermedia* (FGSC #2559) and *N. crassa* as described by Shahi et al.[20]. For this purpose, the *N. crassa* standard strain (74-OR23-1VA) was transformed by the NTC (nourseothricin)-resistance gene and used for subsequent screening. An equal number of protoplasts (~$2 \times 10^5$) from the two fungal strains were fused with the aid of PEG/CaCl₂. The protoplast fusants were grown regenerated on the regeneration media for 1 day and subsequently on overlaid top agar containing 30 μg/ml NTC for 2–3 days for screening the *N. crassa* recipient. Protoplast regenerants resistant to NTC were transferred into PDA plates containing 30 μg/ml NTC and incubated for 1 day before detecting NiFV1. After confirming the presence of NiFV1 by the one-step colony PCR method[71,80], NiFV1-positive colonies were anastomosed with the original nontransformed recipient *N. crassa* strain. A mycelial plug taken from the recipient side was again anastomosed with the original recipient. This hyphal fusion step was repeated three times.

The virus-infected *N. crassa helper-5* strain (FGSC #8747; Δ*mat his-3 tk⁺ hph cyh-1, Bml pan-2*) was created as a donor strain for virus horizontal transfer to a series of recipient strains. Specifically, the virus-free *helper-5* strain and the virus-infected *N. crassa* wild strain (FGSC #10950, NcFV1; FGSC #3975, NcPV1) were cocultured into a slant of Vogel's sucrose medium containing 10 μg/ml pantothenic acid and 25 μg/ml histidine. Subsequently, the virus-infected heterokaryon was forced by a passage into a slant of minimal medium containing 1.5 μg/ml benomyl and then the virus-infected *helper-5* strain was purified through microconidia preparation on SC medium containing 1 mM iodoacetate. Similarly, the standard wild-type strain (74-OR23-1VA) and the derived KO and KI strains were infected by forcing heterokaryon with the virus-infected *helper-5* strain and then were extracted by spreading onto a plate of Vogel's FGS medium containing 5 μM FUDR (5-fluorodeoxyuridine) and 1 mM uracil after microconidia preparation. A schematic diagram for the manipulation was described in Supplementary Fig. 2.

**RNA analyses.** The total RNA and dsRNA fractions were modified by the method of Eusebio-Cope and Suzuki[48]. Specifically, *Neurospora* strains were grown with shaking in Vogel's minimal medium N at 32 °C for 2 days and were harvested by filtration. Tissues were transferred into 22-ml screw-capped tubes (Sarstedt, 72.694) containing 0.40.6-mm zirconia beads, 450 μl of 2× STE, 50 μl of 10% SDS, and 300 μl of phenol/chloroform, and were extracted by using the Minilys homogenizer (Bertin Instruments) for 30 s at maximum speed. After centrifugation at 15,000 rpm for 5 min at 4 °C, 500 μl of the upper phase was transferred into a 1.5-ml microtube and 50 μl of 3 M sodium acetate and 400 μl of isopropanol were added. The concentrations of total RNA were measured by using the Qubit RNA Assay Kit (Thermo Fisher), and the qualities were confirmed by agarose gel electrophoresis. For dsRNA purification, equal amounts of total RNA were dissolved in 420 μl of 1× STE after isopropanol precipitation, then incubated in a 65 °C water bath for 15 min, and quickly chilled on ice. Then, 80 μl of ethanol were added as described by Okada et al.[81] to purify dsRNA. Complementary DNA (cDNA) was prepared by ReverTra Ace qPCR RT Master Mix with gDNA Remover (Toyobo), and quantitative PCR was performed three times using THUNDERBIRD Probe qPCR Mix (Toyobo) and a LightCycler 96 system (Roche Diagnostics) with specific primers summarized in Supplementary Table 2. Measurements were performed twice and averaged. Relative amounts of the respective transcripts were shown with the value for the virus-free standard strain as 1.

Dioxigenin-labeled DNA probes for *N. crassa* genes were prepared by genomic PCR and used in northern blotting according to the manufacturer's instructions. See Supplementary Table 3 for the primer sequences.

**In silico detection of viral sequences**. RNA-seq data were obtained from NCBI. After quality trimming by trimommatic[82], the reads were mapped to genomic DNA, rDNA, mtDNA, and tRNA of the standard *N. crassa* strain 74-OR23-1VA with Bowtie2[83] to obtain unmapped reads. Unmapped reads were then assembled de novo with Trans-ABySS (https://www.bcgsc.ca/resources/software/trans-abyss) into contigs. To create local fungal virus database, potential fungus was extracted from current virus database (NCBI: txid10239). The candidates of fungal viral sequences were selected from the contigs by a BlastX search ($E$ value < $1 \times 10^{-3}$) using the local fungal virus database.

**High-throughput sequence analysis of a *N. intermediate* strain**. The dsRNA-enriched fractions were obtained from petri dish-grown mycelia of a strain of *N. intermediate* (FGSC # 2559, H2125-1) as described by Chiba et al.[44] The dsRNA preparation (17.8 ng/μl) was subjected for cDNA library construction using the TruSeq Stranded Total RNA LT Sample Prep (Illumina, San Diego, CA, USA) and then next-generation sequencing in Illumina technology (HiSeq 2000, 100-bp paired-end reads). Raw data for this project were deposited in NCBI Sequence Read Archive (SRA) under accession No. DRR248874. The cDNA library construction and deep-sequencing analysis were performed by Macrogen Japan, Ltd. A total of 56,528,326 paired-end reads (5,709-Mb read sequences) were assembled into 8415 contigs (~7760 nt in length, average 983 nt) using de novo assembler of CLC Genomics Workbench (version 11, CLC Bio-Qiagen). These contigs were subsequently used as queries for a local BLAST search against the RefSeq annotated viral-genome database.

**Chromatin immunoprecipitation (ChIP) assay and western blotting**. We followed the procedures of the ChIP assay and Western blotting as described previously[84]. The following antibodies were used: anti-H3K4me2 antibody (active motif, 39141), anti-RNA polymerase II CTD repeat YSPTSPS (phospho S5) antibody (Abcam, ab5131), anti-alpha-tubulin antibody (Sigma-Aldrich, T6199), and anti-FLAG antibody (MBL, M185-3). RT-qPCR experiments were performed two times using FAST SYBR Green master kit (KAPA) with the listed primers (Supplementary Table 3) and analyzed using a LightCycler® 96 System (Roche Diagnostics). The standard *N. crassa* strain (74-OR23-1VA) was engineered such that *rrp-3-Flag*, *dcl-2-Flag*, or *qde-2-Flag* was knocked in. Alpha-tubulin detected by Western blotting with anti-alpha-tubulin antibody was used as a loading control.

**Differential gene expression (DEG) analysis**. DEG analysis was performed by the standard HISAT-StringTie-Ballgown pipeline[85]. Based on the results of in silico detection of virus-candidate sequences, we choose ten virus-infected strains (SRR089835, SRR089840, SRR797998, SRR798015, SRR798021, SRR798029, SRR798030, SRR798051, SRR798054, and SRR798057) and ten highly likely virus-free strains (SRR797950, SRR797951, SRR797954, SRR797955, SRR797956, SRR797961, SRR797962, SRR797964, SRR797965, and SRR797967) and compared their levels of gene expression.

**Phylogenetic analyses**. The sequences of mycovirus used in this study were obtained from the NCBI website and are summarized in Supplementary Table 4. The RdRP or CP sequences were aligned using the online tool MAFFT (http://mafft.cbrc.jp/alignment/server/index.html) with default parameters[86]. Maximum-likelihood phylogenetic tree analyses were generated using RAxML-NG[87] with 1000 bootstrap replicates and the specific model selected by ModelTest-NG[88], and were visualized using the graphical viewer FigTree (http://tree.bio.ed.ac.uk/software/figtree/). SNPs were identified using the variant calling and core genome alignment program Snippy (https://github.com/tseemann/snippy) with the *N. crassa* wild strain RNA-seq reads and the standard 74-OR23-1VA strain (annotation NC12_fixed) genome assembly. The core SNP alignment was used for phylogenetic analysis as mentioned above.

**Availability of materials and fungal strains**. Commercially unavailable fungal strains and reagents are available from the authors upon reasonable request, through a material transfer agreement with Okayama University or Fukui University.

**Reporting summary**. Further information on research design is available in the Nature Research Reporting Summary linked to this article.

## Data availability
The authors declare that the data supporting the findings of this study are available within the article and its Supplementary Information files, or are available on request. The RNA-seq data generated in this study were deposited in NCBI Sequence Read Archive (SRA) under accession number DRR248874. The complete viral genomic sequences are deposited in DDBJ/EMBL/GenBank under accession numbers LC530174 for NiFV1, LC530175 for NcFV1-JW60, and LC530176 and LC530177 for bisegmented NcPV1-JW35, respectively. The near-complete genomic sequences of other viruses are deposited in DDBJ/EMBL/GenBank under accession numbers LC586022-LC586028 for

seven different strains of the species Neurospora carassa fusarivirus 1 (Supplementary Table 1). Source data are provided with this paper.

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

## Acknowledgements

This study was supported in part by Yomogi Inc. (to N.S.), the Joint Usage/Research Center, Institute of Plant Science and Resources, Okayama University (Nos. 3023, 3123, and R224 to S.H.), Research Grants from the University of Fukui (to S.H.), and Grants-in-Aid for Scientific Research (A and B) and on Innovative Areas, and Grants-in-Aid for Research Activity Start-up from the Japanese Ministry of Education, Culture, Sports, Science, and Technology (MEXT) (KAKENHI 25252011 and 16H06436, 16H06429, and 16K21723 to N.S. and HK, and 19H04828 to S.H.). We are grateful to Drs. Donald L Nuss, Bradley I. Hillman, and Satoko Kanematsu for the generous gift of the fungal/viral strains, and Ms. Sakae Hisano for technical assistance. Our gratitude is extended to Profs. Carlo Cogoni and Caterina Catalanotto who provided several quelling-deficient mutants at an initial stage of the study. We gratefully acknowledge the Neurospora gene knockout consortium (Neurospora Functional Genomics grant #P01GM68087, NIH).

## Author contributions

N.S. initiated the project. N.S. and S.H. conceived the study and designed the experiments. S.H. introduced viruses into mutant fungal strains, and comparatively analyzed them by western blotting, ChIP, and RT-qPCR. A.Y. performed RT-qPCR and dsRNA extraction. A.E.C. and S.S. designed and performed virion transfection and gel electrophoresis of viral dsRNA. S.S. performed protoplast fusion assay. AA obtained the entire virus genome sequences by RACE and carried out northern blotting and dsRNA gel electrophoresis. S.H. and S.M. mined public databases for viral sequences and phylogenetically analyzed them. HK prepared dsRNA and analyzed NGS data of viruses. S.H., S.M., H.K., and N.S. analyzed the data. S.H., H.K., and N.S. wrote and reviewed the paper. All authors discussed the data, read, and approved the paper.

## Competing interests

The authors declare no competing interests.
