## [Peer Review File · Nature Communications]

REVIEWER COMMENTS

Reviewer #1 (Remarks to the Author):

Neurospora crassa has been an important model organism for RNAi research but the antiviral function of the RNAi pathway was not previously demonstrated. In this study, the authors demonstrated that *Neurospora* is a natural viral host by discovering multiple dsRNA viruses in different *Neurospora* isolates. In addition, the authors developed the method to demonstrate that *N. crassa* can be used as an experimental system to examine viral replication and antiviral response. The results showed that the RNAi components Dicer and QDE-2 suppress viral replication in *Neurospora*. Furthermore, viral infection induces RNAi components transcriptionally. Together, these results represent an important advance in fungal antiviral research by laying the foundation in a model organism. Although the study was well carried out, there are a few issues the author should address before publication.

1. Since a major question in this study concerns the role of the *Neurospora* RNAi pathway in the antiviral response, more complete overview of the known *Neurospora* pathway is warranted in the introduction. In addition, the current study is also highly related to a previous study (Reference 37) which demonstrated the existence of dsRNA-induced transcriptional response, including many RNAi components and putative antiviral genes. This earlier study suggested that this response is a host defense response for viral infection. Moreover, more coverage of the relevant studies in *C. parasitica* is also needed.
2. Since this study established the system to test the role of potential antiviral genes in *Neurospora*, the authors probably can check whether some of the putative antiviral genes (Mx, RNA helicase and others) previously identified in reference 37 are involved in antiviral response. It is not necessary to examine many of them, but even if the authors can demonstrate the involvement of just one of the novel antiviral genes will greatly strengthen this study by providing novel mechanistic insights of the antiviral response..
3. Line 129: Why NiFV1 dsRNA accumulated much less in *N. crassa* than in *N. intermedia*?
4. Line 138-140: It will be nice to provide more background on why a helper-5 strain is needed.
5. Line 160 and Figure 2: Why the accumulation of NcPV1 in the RNAi mutants was much lower than those for NcFV1?
6. Figure 3c, more explanation is needed for this result in the text. Did the authors check the induction of other DRAGs previously identified in reference 37?
7. Figure 3e and line 188: It was previously shown that dsRNA expression (37) or DNA damage (which also induced dsRNA from rDNA loci) (Lee et al, 2009) can induced the endogenous QDE-2 levels. It is surprising that the tagged-QDE2 level was not increased and even decreased after viral infection. Can the comparison of NcFV1 and NcPV1 viral sequences reveal anything interesting based on what was previously known viral RNAi suppressors?
8. Figure 4: The dsRNA-induced transcriptional pathway should be incorporated into the figure.

Reviewer #2 (Remarks to the Author):

The major claim of the paper is the establishment of a system to study virus-fungus interaction in the model fungus *Neurospora crassa*, and its use to determine the fungal genes involved in antiviral defense in *N. crassa*. The authors showed also for the first time the presence of different type viruses in three *Neurospora* species, and the replication of viruses of *Rosellinia necatrix* and *Chryphonectria parasitica* in *N. crassa*. The authors showed results that will be of interest to others working with mycoviruses and fungus-virus interactions. The results obtained in this work were compared by the authors with previous results obtained in the fungus *C. parasitica*, that has been extensively used as model to study virus-fungus interactions using mycoviruses of *C. parasitica* or

of other fungus as *Rosellinia necatrix*.

The level of detail provided in Methods will allow to reproduce some experiments, but for instance for the preparation of protoplasts or for protoplast fusion will be necessary to consult other papers. For a better understanding of the experimental mycoviruses introduction, the authors could draw an explanatory scheme to clarify assays described in materials and methods and results sections.

Some specific comments:

NcFV1 A type was used for this study, explain the difference with type B and C, and the interest of using A type instead other type.

Indicate the natural host of Mycoreovirus 1.

In this work it is shown that *Neurospora* spp. are naturally infected by partitivirus, even if they are infected with Betapartitivirus, the authors should explain why is interesting to use *Rosellinia necatrix* partitivirus 2 instead another type of virus, to increase the diversity of studied viruses. Some experiments included RnPV2, but not all of them, the inclusion of this mycovirus in all the assays would increase the relevance of the manuscript, since the fungus could be proposed as a general model system for the study of mycovirus-fungus interaction.

The authors showed the dsRNA accumulation of RnPV2 and MyRV1, but another method should be used to demonstrate the replication of both viruses in *N. crassa* and if this replication is stable.

According to Extended Data Table 1, *Neurospora* spp. can also host viruses of the family *Togaviridae*. Include in the list of viral families in the text.

In the third section of results, the conclusion is that the two Dicers DCL-1/2 and one Argonaut QDE-2 play a major role in anti-viral RNA silencing, but according with the results this is true for the fusarivirus. For the partitivirus, even when the differences are no significant between wt and mutants, DCL-1/2 has no role in anti-viral silencing, only in (Dqde-2/Dsms-2) and (Dqde-1/DSad-1/Drrp-3) there is an increase of NcPV1 accumulation. The partitivirus is a dsRNA virus, and the fusarivirus is a ssRNA (+) virus, and dsRNA accumulation has been used to determine the effect of the mutation of genes involved in silencing pathways, the results are clear, but maybe a northern blot would be a complementary analysis for this purpose.

The Northern blot in Figure 3a has a strong background and it is difficult to see the bands, maybe the authors should substitute the photo in the figure.

Figure 3c shows different gene expression, but this analysis was performed using "12 possible virus-infected strains and 12 highly likely virus-free strains" (pag. 17 in material and methods), if the authors can no assure that the fungal strains are or no infected with mycoviruses, this analysis should not be considered in the result section.

In the discussion section the authors summarized the results and only discussed with other studies developed with in *C. parasitica*, but another studies have been published about the roles of argonautes and dicers, for instance in *Sclerotinia sclerotiorum*, that maybe could be mentioned.

Reviewer #3 (Remarks to the Author):

The study shows that several double stranded RNA viruses and newly identified ss(+)RA viruses can be transfected to *Neurospora crassa*. The authors also show that RNAi genes, DCL-1/2 and QDE-2, were induced upon virus infection. One argonaute mutant, Δ qde-2, had an increase in accumulation of NcFV1. Probed by NcFV1, they also showed that the two dicer genes work redundantly in antiviral defense, and that the other RNAi genes are not playing redundant roles in antiviral RNAi. The RNA-Seq data showed differential expression of dcl-2, qde-2, and rrp-3 genes in the virus infected wild-type strain, as well as the corresponding changes in protein accumulation in a virus-specific manner. The transfections were achieved by multiple methods, indicating that *N. crassa* will be a very good model system to study virus-host interactions and co-infections of ds- and ss(+) viral genome types. Additionally, the manuscript presented the evidence to support the model of antiviral RNA silencing pathway.

The manuscript contains extensive data/analysis and provides novel insights into virus-host interaction in a model system, which sets the foundation for a lot of in depth investigation on a

genetic system with ease, such as potential RNA silencing suppressor targeting argonaute to induce autophagy, as eluded by the authors. The system can also be applicable to understanding other agricultural important fungal pathogens as hosts of viruses, especially for fungi belonging to Ascomycetes. I think the authors can elaborate more on the use of *N. crassa* as a model system and how specific pathogen control or antiviral strategies can be devised. Overall, the last paragraph of discussion can be an opportunity to emphasize the importance of such a system and should be strengthened.

Suggestions on minor edits:

L30: should be "one gene-one enzyme"

L32: probably "because of unconfirmed virus infection despite numerous" would sound better

L47~48: two "generally" in one sentence: maybe change to "commonly" for one of them

L50: delete "relatively" to make it sound better

L53: add hyphen to "fusion-based"

L107: belong into should be "belong to"

REVIEWER COMMENTS

Reviewer #1 (Remarks to the Author):

Neurospora crassa has been an important model organism for RNAi research but the antiviral function of the RNAi pathway was not previously demonstrated. In this study, the authors demonstrated that *Neurospora* is a natural viral host by discovering multiple dsRNA viruses in different *Neurospora* isolates. In addition, the authors developed the method to demonstrate that *N. crassa* can be used as an experimental system to examine viral replication and antiviral response. The results showed that the RNAi components Dicer and QDE-2 suppress viral replication in *Neurospora*. Furthermore, viral infection induces RNAi components transcriptionally. Together, these results represent an important advance in fungal antiviral research by laying the foundation in a model organism. Although the study was well carried out, there are a few issues the author should address before publication.

Response: Thank you for the reviewer's overall positive comments on our work. We must confess that our descriptions of most subdivisions are limited due largely to length limitations set by [redacted] which we originally submitted our paper to. We have added explanations to some methods and expanded the Results and Discussion sections.

1. Since a major question in this study concerns the role of the *Neurospora* RNAi pathway in the antiviral response, more complete overview of the known *Neurospora* pathway is warranted in the introduction. In addition, the current study is also highly related to a previous study (Reference 37) which demonstrated the existence of dsRNA-induced transcriptional response, including many RNAi components and putative antiviral genes. This earlier study suggested that this response is a host defense response for viral infection. Moreover, more coverage of the relevant studies in *C. parasitica* is also needed.

Response: The Introduction and associated sections have greatly been expanded to introduce the *N. crassa* RNA silencing pathways and the *C. parasitica* antiviral RNA silencing (page 3, 2nd paragraph).

2. Since this study established the system to test the role of potential antiviral genes in *Neurospora*, the authors probably can check whether some of the putative antiviral genes (Mx, RNA helicase and others) previously identified in reference 37 are involved in antiviral response. It is not necessary to examine many of them, but even if the authors can demonstrate the involvement of just one of the novel antiviral genes will greatly strengthen this study by providing novel mechanistic insights of the antiviral response.

Response: Thank you for this suggestion. We have added data on Hex1 (page 9, Supplementary Fig. 6) whose orthologue in *Fusarium graminearum* was previously identified as a host factor for the replication and symptom induction of a fusarivirus of (Son et al., JVI, 2013). [redacted].

3. Line 129: Why NiFV1 dsRNA accumulated much less in *N. crassa* than in *N. intermedia*?

Response: We assume that the less NiFV1 dsRNA accumulation in *N. crassa* may be explained by the greater adaptability of NiFV1 to *N. intermedia*. As mentioned on page 6, similar phenomena were previously observed after virus introduction into new experimental hosts.

4. Line 138-140: It will be nice to provide more background on why a helper-5 strain is needed.

Response: More explanation has been added in the main text and legend of a supplementary figure (Fig. 4) illustrating an experimental procedure for horizontal virus transfer using the helper-5 strain.

5. Line 160 and Figure 2: Why the accumulation of NcPV1 in the RNAi mutants was much lower than those for NcFV1?

Response: Please note that these viruses belong to totally different virus families: Partitiviridae and proposed Fusariviridae. Different members even within the same Partitiviridae show different susceptibility to antiviral RNA silencing (Chiba et al., JVI, 2013; Virus Res, 2016; Aulia et al., Curr Res Virol Sci, 2020). Some viruses accumulate much more in RNA silencing-deficient mutants than in wild-type strains, whereas other do not. The current virology cannot explain this phenomenon well.

6. Figure 3c, more explanation is needed for this result in the text. Did the authors check the induction of other DRAGs previously identified in reference 37?

Response: Other dsRNA-induced genes such as NCU04490 (6-16 family), NCU07036 (3'-5' exonuclease), NCU04472 (RNA helicase), NCU09495 (set-6), and NCU00947 (unknown function) have also been confirmed to be induced. This has briefly been touched in the text on page 8. We would rather not include these data in the current manuscript, but report them together with deep virological data in the near future.

7. Figure 3e and line 188: It was previously shown that dsRNA expression (37) or DNA damage (which also induced dsRNA from rDNA loci) (Lee et al, 2009) can induced the endogenous QDE-2 levels. It is surprising that the tagged-QDE2 level was not increased and even decreased after viral infection. Can the comparison of NcFV1 and NcPV1 viral sequences reveal anything interesting based on what was previously known viral RNAi suppressors?

Response: this is a very interesting question that we next attempt to answer in one of the follow-up papers. Below is what we speculate. RNA silencing suppressors have been identified only from two hypoviruses (Segers et al., 2006; Aulia et al., in preparation), a fusarivirus (Yu et al., MPP, 2020) and one mycovirus (Yaegashi et al., Virology, 2013) thus far. We assume that a NcFV1 suppressor leads to QDE-2 degradation likely via an autophagy or a proteasome-mediated pathway similar to that reported for an RNA silencing suppressor of plant virus origin (Michaeli et al., PNAS, 2019).

8. Figure 4: The dsRNA-induced transcriptional pathway should be incorporated into the figure.

Response: Incorporated in new Fig. 4.

Reviewer #2 (Remarks to the Author):

The major claim of the paper is the establishment of a system to study virus-fungus interaction in the model fungus *Neurospora crassa*, and its use to determine the fungal genes involved in antiviral defense in *N. crassa*. The authors showed also for the first time the presence of different type viruses in three *Neurospora* species, and the replication of viruses of *Rosellinia necatrix* and *Chryphonectria parasitica* in *N. crassa*. The authors showed results that will be of interest to others working with mycoviruses and fungus-virus interactions. The results obtained in this work were compared by the authors with previous results obtained in the fungus *C. parasitica*, that has been extensively used as model to study virus-fungus interactions using mycoviruses of *C. parasitica* or of other fungus as *Rosellinia necatrix*.

The level of detail provided in Methods will allow to reproduce some experiments, but for instance for the preparation of protoplasts or for protoplast fusion will be necessary to consult other papers. For a better understanding of the experimental mycoviruses introduction, the authors could draw an explanatory scheme to clarify assays described in materials and methods and results sections.

Response: The reviewer's positive evaluation of our study is appreciated. The detailed procedures have been described for protoplast fusion (page 12) and illustrated for helper-5 mediated horizontal virus transfer (Supplementary Fig. 4).

Some specific comments:

NcFV1 A type was used for this study, explain the difference with type B and C, and the interest of using A type instead other type.

Response: NcFV1 subgroups A to C can be differentiated by their phylogenetic relationships and sequence identity. This has been added on page 3.

Indicate the natural host of Mycoreovirus 1.

Response: Original host of MyRV1 has been described on page 5.

In this work it is shown that *Neurospora* spp. are naturally infected by partitivirus, even if they are infected with Betapartitivirus, the authors should explain why is interesting to use *Rosellinia necatrix* partitivirus 2 instead another type of virus, to increase the diversity of studied viruses. Some experiments included RnPV2, but not all of them, the inclusion of this mycovirus in all the assays would increase the relevance of the manuscript, since the fungus could be proposed as a general model system for the study of mycovirus-fungus interaction.

Response: An explanation has been added as to why RnPV2 was used. Chiba et al. (PLoS Pathogens, 2011) showed a plant partitivirus closely related to RnPV2 to have been endogenized into the host nuclear genome. Thus, we anticipated that RnPV2 could be transfectable to *N. crassa*. RnPV2 has been included in the analyses in parallel with NcFV1 and NcPV1 (see new Figs. 2 and 3).

The authors showed the dsRNA accumulation of RnPV2 and MyRV1, but another method should be used to demonstrate the replication of both viruses in *N. crassa* and if this replication is stable.

Response: We are confident about the replication of RnPV2 and MyRV1 in *N. crassa*. DsRNA gel analysis is one of the most reliable methods for RNA virus replication. We confirmed their replication not only in wild type *N. crassa* but also in mutant *N. crassa* strains (see Figs. 2 & 3 and Supplementary Figs. 5 & 6 for RnPV2). As mentioned in the text (page 5), MyRV1 was very unstable and was not used in subsequent analyses. Such an unstable infection by a mycovirus is observed even in its original fungal host (Aulia et al., Virology, 2019).

According to Extended Data Table 1, *Neurospora* spp. can also host viruses of the family Togaviridae. Include in the list of viral families in the text.

Response: Thank you for this suggestion. We feel that it is too early to conclude based solely on the partial genomic sequence that the detected toga-like virus is a member of the family Togaviridae. The partially sequenced toga-like mycovirus has much shorter genomic elements of <4.0 kb compared to a typical togavirus with a genome of ~10 kb. It is unknown whether the toga-like mycovirus is multi-segmented, or whether it forms spherical particles like animal togaviruses. Therefore, it is safe to wait until the full-genome sequence of the toga-like virus from *N. crassa* strain D23 is determined. Taking these into account, "alpha-like supergroup," broadly including toga-like, virga-like, and other related viruses, has been added in the text (page 6, Supplementary Table 1).

In the third section of results, the conclusion is that the two Dicers DCL-1/2 and one Argonaut QDE-2 play a major role in anti-viral RNA silencing, but according with the results this is true for the fusarivirus. For the partitivirus, even when the differences are no significant between wt and mutants, DCL-1/2 has no role in anti-viral silencing, only in (Dqde-2/Dsms-2) and (Dqde-1/DSad-1/Drrp-3) there is an increase of NcPV1 accumulation. The partitivirus is a dsRNA virus, and the fusarivirus is a ssRNA (+) virus, and dsRNA accumulation has been used to determine the effect of the mutation of genes involved in silencing pathways, the results are clear, but maybe a northern blot would be a complementary analysis for this purpose.

Response: RT-qPCR has been performed on (+)ssRNA of all the three viruses, NcFV1, NcPV1, and RnPV2. The results have been shown in Supplementary Fig. 5 which has a trend similar to their dsRNA accumulation. Whereas over 25-fold elevation of NcFV1 (+)ssRNA was observed in $\Delta dcl-1/2$, $\Delta qde-2$ and $\Delta qde-2/\Delta sms-2$, no over 4-fold augmentation of the partitivirus (+)ssRNA was observed in any mutant. These observations have been described and interpreted on pages 6 and 11.

The Northern blot in Figure 3a has a strong background and it is difficult to see the bands, maybe the authors should substitute the photo in the figure.

Response: The northern blot panel has been replaced with that of a brighter version.

Figure 3c shows different gene expression, but this analysis was performed using “12 possible virus-infected strains and 12 highly likely virus-free strains” (pag. 17 in material and methods), if the authors can no assure that the fungal strains are or no infected with mycoviruses, this analysis should not be considered in the result section.

Response: We have deleted two *N. crassa* strains that were not confirmed to be dsRNA positive by our analyses. Thus, Fig. 3c now has included the data of 10 fungal strains and shown similar induction patters to the previous data.

In the discussion section the authors summarized the results and only discussed with other studies developed with in *C. parasitica*, but another studies have been published about the roles of argonautes and dicers, for instance in *Sclerotinia sclerotiorum*, that maybe could be mentioned.

Response: The discussion section has been expanded to include discussion on the RNA silencing pathways in other filamentous fungi such as *S. sclerotiorum* and *F. graminearum* (page 10).

Reviewer #3 (Remarks to the Author):

The study shows that several double stranded RNA viruses and newly identified ss(+)RA viruses can be transfected to *Neurospora crassa*. The authors also show that RNAi genes, DCL-1/2 and QDE-2, were induced upon virus infection. One argonaute mutant, $\Delta qde-2$, had an increase in accumulation of NcFV1. Probed by NcFV1, they also showed that the two dicer genes work redundantly in antiviral defense, and that the other RNAi genes are not playing redundant roles in antiviral RNAi. The RNA-Seq data showed differential expression of *dcl-2*, *qde-2*, and *rrp-3* genes in the virus infected wild-type strain, as well as the corresponding changes in protein accumulation in a virus-specific manner. The transfections were achieved by multiple methods, indicating that *N. crassa* will be a very good model system to study virus-host interactions and co-infections of ds- and ss(+) viral genome types. Additionally, the manuscript presented the evidence to support the model of antiviral RNA silencing pathway.

The manuscript contains extensive data/analysis and provides novel insights into virus-host interaction in a model system, which sets the foundation for a lot of in depth investigation on a genetic system with ease, such as potential RNA silencing suppressor targeting argonaute to induce autophagy, as eluded by the authors. The system can also be applicable to understanding other agricultural important fungal pathogens as hosts of viruses, especially for fungi belonging to Ascomycetes. I think the authors can elaborate more on the use of *N. crassa* as a model system and how specific pathogen control or antiviral strategies can be devised. Overall, the last paragraph of discussion can be an opportunity to emphasize the importance of such a system and should be strengthened.

Suggestions on minor edits:

L30: should be “one gene-one enzyme”

L32: probably “because of unconfirmed virus infection despite numerous” would sound better

L47~48: two “generally” in one sentence: maybe change to “commonly” for one of them

L50: delete “relatively” to make it sound better

L53: add hyphen to “fusion-based”

L107: belong into should be “belong to”

Response: All of the suggestions have been incorporated. The Discussion has been expanded to discuss the use of *N. crassa*, as suggested page 11.

REVIEWERS' COMMENTS

Reviewer #1 (Remarks to the Author):

The revised manuscript has addressed by concerns and I support its publication in NC after incorporating the minor changes suggested below.

1. RNAi is most commonly used to describe small RNA-mediated gene silencing mechanisms. I think it's better and clearer to readers if "RNAi" rather than "RNA silencing" was used to describe the silencing mechanism in the paper.
2. In some places, "quelling" was used to describe RNAi during vegetative stage. Although related, quelling originally refers to gene silencing triggered by transgenes or repetitive DNA sequences (see Zhang et al., 2013, *Genes & Dev*, Yang et al, 2015 *Genes & Dev*) while RNAi refers to gene silencing mediated by dsRNA and sRNAs. Please revise in the text.
3. The paper will benefit from English editing from the copy editor after its acceptance.
4. Line 78: "its antiviral activity in *N. crassa* was uncovered". Change to, "its role in antiviral response has not been demonstrated in *N. crassa*".
5. Line 79: "Choudhary et al. demonstrated the transcriptional induction of RNA silencing pathway by transgenic expression of dsRNA, suggesting the existence of RNA silencing defense in this fungus." Change to "Choudhary et al. demonstrated the transcriptional induction of RNAi pathway and other putative antiviral genes by transgenic expression of dsRNA, suggesting that the RNAi pathway may act as part of the viral defense mechanism in this fungus".
6. Line 92: "Collectively, this study represents a foundation for the study of virology in the model organism *N. crassa*.". Replace "represents" by "establishes".
7. Line 228: "Many genes including RNA silencing-related genes were previously reported to be induced at the transcription level by dsRNA expression." Change to "Many genes including those of the RNAi pathway and putative antiviral response were previously reported to be induced at the transcription level by dsRNA expression in *Neurospora*".
8. Line 237: "inducive" to "inducible".Line 237-240: "Some of the other genes identified as dsRNA inducive genes by Choudhary et al.³⁵ were also up-regulated upon virus infection, which included NCU04490 (6-16 family), NCU07036 (3'-5' exonuclease), NCU04472 (RNA helicase), NCU09495 (set-6), and NCU00947 (unknown function). These results should be shown in the paper.
9. Line 292: "*N. crassa* is the first organism to be used for genetic dissection of the RNA silencing pathway. There are two types of RNA silencing: quelling and MSUD (meiotic silencing by unpaired DNA)." Change to "*N. crassa* is the first organism to be used for genetic dissection of the RNAi pathway in fungi.". Add references here.

Reviewer #2 (Remarks to the Author):

The authors have included all modifications I have proposed, and in general, after the changes proposed by all reviewers, the manuscript have been improved. My last suggestion is that the authors review the manuscript to eliminate typo errors, there are some of them in the main text and in the supplementary material. Also review the references, especially in Methods, to assure that all of them are correct, because it is possible that some references may not be correct.

REVIEWER COMMENTS

Reviewer #1 (Remarks to the Author):

The revised manuscript has addressed by concerns and I support its publication in NC after incorporating the minor changes suggested below.

1. RNAi is most commonly used to describe small RNA-mediated gene silencing mechanisms. I think it's better and clearer to readers if "RNAi" rather than "RNA silencing" was used to describe the silencing mechanism in the paper.

2. In some places, "quelling" was used to describe RNAi during vegetative stage. Although related, quelling originally refers to gene silencing triggered by transgenes or repetitive DNA sequences (see Zhang et al., 2013, *Genes & Dev*, Yang et al, 2015 *Genes & Dev*) while RNAi refers to gene silencing mediated by dsRNA and sRNAs. Please revise in the text.

Response: I agree with the reviewer that "quelling" was discovered by the Italian research group led by Macino and originally referred to transgene-mediated gene silencing. However, "quelling" was later analyzed genetically and mechanistically and turned out to have the same pathway to silence target genes as RNAi. The bottom line here is that there are several ways by which dsRNA is generated as shown in new Fig. 6. The above-mentioned papers have been cited on page 9 (see Response to point 9).

3. The paper will benefit from English editing from the copy editor after its acceptance.

4. Line 78: "its antiviral activity in *N. crassa* was uncovered". Change to, "its role in antiviral response has not been demonstrated in *N. crassa*".

Response: Done.

5. Line 79: "Choudhary et al. demonstrated the transcriptional induction of RNA silencing pathway by transgenic expression of dsRNA, suggesting the existence of RNA silencing defense in this fungus." Change to "Choudhary et al. demonstrated the transcriptional induction of RNAi pathway and other putative antiviral genes by transgenic expression of dsRNA, suggesting that the RNAi pathway may act as part of the viral defense mechanism in this fungus".

Response: Done.

6. Line 92: "Collectively, this study represents a foundation for the study of virology in the model organism *N. crassa*". Replace "represents" by "establishes".

Response: Done.

7. Line 228: "Many genes including RNA silencing-related genes were previously reported to be induced at the transcription level by dsRNA expression." Change to "Many genes including those of the RNAi pathway and putative antiviral response were previously reported to be induced at the transcription level by dsRNA expression in *Neurospora*".

Response: Done.

8. Line 237: "inducive" to "inducible". Line 237-240: "Some of the other genes identified as dsRNA inducive genes by Choudhary et al.35 were also up-regulated upon virus infection, which included NCU04490 (6-16 family), NCU07036 (3'-5' exonuclease), NCU04472 (RNA helicase), NCU09495 (set-6), and NCU00947 (unknown function). These results should be shown in the paper.

Response: The data have been included in new supplementary Fig. 5.

9. Line 292: "N. crassa is the first organism to be used for genetic dissection of the RNA silencing pathway. There are two types of RNA silencing: quelling and MSUD (meiotic silencing by unpaired

DNA).” Change to “*N. crassa* is the first organism to be used for genetic dissection of the RNAi pathway in fungi.” Add references here.

Response: Appropriate references have been added.

Reviewer #2 (Remarks to the Author):

The authors have included all modifications I have proposed, and in general, after the changes proposed by all reviewers, the manuscript have been improved. My last suggestion is that the authors review the manuscript to eliminate typo errors, there are some of them in the main text and in the supplementary material. Also review the references, especially in Methods, to assure that all of them are correct, because it is possible that some references may not be correct.

Response: The manuscript has carefully been reviewed.